# Synthesis, Characterization, DFT and Photocatalytic Studies of a New Pyrazine Cadmium(II) Tetrakis(4-methoxy-phenyl)-porphyrin Compound

**DOI:** 10.3390/molecules27123833

**Published:** 2022-06-14

**Authors:** Chadlia Mchiri, Louis-Charl C. Coetzee, Faycal Chandoul, Abdesslem Jedidi, Adedapo S. Adeyinka, Nomampondo Magwa, Thierry Roisnel, Sana Ben Moussa, Habib Nasri

**Affiliations:** 1Laboratoire de Physico-Chimie des Matériaux (LPCM), Faculté des Sciences de Monastir, Avenue de l’environnement, University of Monastir, Monastir 5019, Tunisia; habibnasri398@gmail.com; 2Department of Chemistry, University of South Africa, Private Bag X6, Florida, Roodepoort, Johannesburg 1710, Gauteng, South Africa; louischarlc0@gmail.com (L.-C.C.C.); mnomampondo@yahoo.com (N.M.); 3Faculty of Mathematical, Physical and Natural Sciences of Tunis, University of Tunis El Manar, Tunis 2092, Tunisia; faycal.chandoul@fst.utm.tn; 4Department of Chemistry, Faculty of Science, King Abdulaziz University, Jeddah 21589, Saudi Arabia; ajedidi@kau.edu.sa; 5Research Centre for Synthesis and Catalysis, Department of Chemical Sciences, University of Johannesburg, Auckland Park, Johannesburg 2006, Gauteng, South Africa; 6Centre de Diffractométrie X, Institut des Sciences Chimiques de Rennes, UMR 6226, CNRS–Université de Rennes 1, Campus de Beaulieu, 35042 Rennes, France; thierry.roisnel@univ-rennes1.fr; 7Faculty of Science and Arts, Mohail Asser, King Khalid University, Abha 61421, Saudi Arabia; sanabm2@hotmail.fr

**Keywords:** cadmium porphyrin, molecular structure, DFT-calculations, NBO analysis, photocatalytic activity, non-covalent interactions (NCI)

## Abstract

This study describes the synthesis, theoretical investigations, and photocatalytic degradational properties of a new (pyrazine)(meso-tetrakis(4-tert-methoxyphenyl)-porphyrinato)-cadmium (II) ([Cd(TMPP)-Pyz]) complex (**1**). The new penta-coordinated Cd^II^ porphyrin complex (**1**) was characterized by various spectroscopic techniques, including FT-IR, NMR, UV-visible absorption, fluorescence emission, and singlet oxygen, while its molecular structure was studied using single crystal X-ray diffraction. The UV–Vis spectroscopic study highlighted the redshift of the absorption bands after the insertion of the Cd(II) metal ion into the TMPP ring. The co-coordination of the pyrazine axial ligand enhanced this effect. A fluorescence emission spectroscopic study showed a significant blueshift in the Q bands, accompanied by a decrease in the fluorescence emission intensity and quantum yields of Φ_f_ = 0.084, Φ_f_ = 0.06 and Φ_f_ = 0.03 for H_2_-TMPP free-base porphyrin, [Cd(TMPP)] and [Cd(TMPP)(Pyz)] (**1**) respectively. Singlet oxygen revealed that the H_2_-TMPP porphyrin produced the most efficient singlet oxygen quantum yield of (Φ_Δ_ = 0.73) compared to [CdTMPP] (Φ_Δ_ = 0.57) and [Cd(TMPP)(Pyz)] (**1**) (Φ_Δ_ = 0.13). In the crystal lattice, the [Cd(TMPP)Pyz] was stabilized through non-covalent intermolecular interactions (NCI), such as the hydrogen bonds C-H···N and C-H···Cg. Additionally, crystal explorer software was then utilized to measure the quantitative analysis of the intermolecular interactions in the unit cell of the crystal structure and established that the C-H···π interaction dominated. The Natural bond orbital (NBO) analysis revealed that each molecule is stabilized by hyperconjugation and charge delocalization. As a photocatalyst, the coordination complex **1** showed excellent photocatalytic activity toward the degradation of Levafix Blue CA reactive dye (i.e., dye photo-degradation of 80%).

## 1. Introduction

Porphyrins and their derivatives are highly π-conjugated heterocyclic molecules. Due to their extreme importance among natural compounds, they exist in nature and are called the pigments of life [1]. Although they can be extracted from plant matter, the amount obtained is minimal. Thus, synthesizing them will increase their yield and diversify their structures, characteristics and functions, resulting in a wide range of potential applications, which includes recyclable absorbents for selective separation of methyl blue [2], dye adsorption [3], and photoconductivity of a framework [4].

Their structural diversities with core-, meso-, and β-modifications play a significant role in donating electrons, suggesting a way to tune their electronic structures and properties as dye sensitizers in DSSCs [5,6]. For example, 5-(4-carboxyphenyl)-10,15, 20-tritolylporphyrin (H_2_TC_1_PP) and 5,10,15,20-tetrakis(4-carboxy phenyl)porphyrin (H_2_TC_4_PP) have been used as photosensitizers in TiO_2_ DSSCs and investigated as effectual photosensitizers for solar-energy harvesting [7]. Additionally, porphyrins have important potential applications in cancer treatment, and numerous porphyrin-based inorganic nanoparticles (NPs) have been designed and applied in cancer therapy [8].

Metal complexes with porphyrin ligands are an attractive class of compounds used as photocatalysts under visible light irradiation [9]. For instance, the photocatalytic activities of tin porphyrins were tested on the photodegradation of orange 7 and 4-chlorophenol in water under visible irradiation, demonstrating successful photocatalytic properties with good performance [10]. Meanwhile, porphyrins have an excellent capacity to produce reactive oxygen species (ROS), including ^1^O_2_, O_2_^•−^ and •OH, for photochemical degradation of organic pollutants [11]. On the other hand, dye pollutants are becoming a significant source of environmental contamination, which is considered a significant threat to the environment. Additionally, serious health problems, such as pathogenic mutation and cancer, can be caused by some of the dyes [12,13,14]. Photodegradation of dye pollutants has attracted more attention for their efficient ability to degrade these pollutants into less dangerous compounds. The promotion of photoreaction in the presence of a catalyst is called heterogeneous photocatalysis. The latter is an emerging technique and is valuable for water and air purification and remediation. In heterogeneous photocatalysis, the photocatalyst is present as a solid, with the reactions taking place at the interface between phases, i.e., solid–liquid or solid–gas [15].

This technology is a low-cost and sustainable option for treating organic and inorganic pollutants present in water [16]. It is worth noting that new applications of metalloporphyrins as adsorbents and photocatalysts for the degradation of organic dyes have emerged in recent years [17]. Thus, the synthesis of new photocatalysts is a popular research topic for the degradation of those organic and inorganic pollutants in water [18].

In the present work, we report the synthesis of a novel (Pyrazine)(meso-tetrakis(4-tert-methoxyphenyl)-porphyrinato)-cadmium(II) ([Cd(TMPP)(Pyz)] (**1**) (Figure 1). This complex (**1**) is obtained with a yield of around 85%. The structural characterization (using ^1^H NMR, FT-IR spectroscopy, and single crystal X-ray diffraction), and the photophysical properties of (**1**) were investigated. Additionally, the geometry of this novel complex **1** was optimized via DFT using a B3LYP/6-31G(d) level of theory, while its intermolecular interactions were quantitatively analyzed at the B3LYP/ DGDZVP level of theory, and natural bond orbital (NBO) charges were simulated to study its electronic properties. Moreover, the photocatalytic performance of [Cd(TMPP)(Pyz)] (**1**) was examined through the photodegradation of Levafix Blue CA reactive dye, which is widely used in the textile industry [19,20]. Lastly, the photodegradation process of the organic pollutant was investigated in water under sunlight irradiation, and a mechanistic pathway for photodegradation was proposed.

## 2. Results

### 2.1. Crystal Structure of [Cd(TMPP)(Pyz)] (1)

The prepared [Cd(TMPP)(Pyz)] complex (**1**) crystallizes in the monoclinic system and P21/c space group, with lattice parameter a = 23.0965(10) Å, b = 17.6690(7) Å, c = 12.4340(5) Å, α = γ = 90°, β = 99.551(2)°, V = 5003.9(4) Å3, Mr = 925.30, Z = 4. The crystal of (**1**) consists of one molecule in the asymmetric unit. The crystal structure of (**1**) is shown in Figure 1, and crystallographic information is listed in Appendix A. In addition, Appendix A shows selected values of experimental and theoretical bond length (Å) and angles (degree, °).

In our structure, the Cd(II) cation is located at the center of the porphyrin ring with Cd-Npyrrole distances of (Cd-Np) of 2.188(17) Å and the nitrogen N5 atom of the pyrazine ligand in the apical position to form a five-coordinate complex with a distorted square pyramidal geometry (Figure 2). The Cd-N(pyridyl) bond length value is 2.3696(19) Å (Table 1).

It is observed that the Cd(II) ion in the center of the complex is located 0.66 Å out-of-plane formed by the four nitrogen atoms of the pyrrole moieties, forming a domed structure on complex **1** (see Figure 2 and Appendix A). Furthermore, the distance made by the Cd(II) atom out of the 24-atom of the macrocycle mean plane (Cd-Ct) for (**1**) was 0.708 Å, which was close to [Cd(TClPP)(py)] with (Cd-Ct) value of 0.729 Å; this distance was shorter than that of [Cd(TBPP)(2-MeHIm)] (Cd-Ct) 0.860 Å (Table 1).

Generally, porphyrins are planar compounds, but due to different factors, the distortion of the macrocycle can also be observed. One of these factors include the metalation of the porphyrin cavity. Additionally, the deformation could be related to the substitution of the macrocycle at beta or meso positions by bulky groups. Scheidt and Lee have identified four types of distortions, which includes dome, saddle, ruffle, and wave [27]. For complex (**1**), the distortion of the macrocycle was studied. Appendix A shows that the [Cd(TMPP)(Pyz)] (**1**) displayed an important distortion (doming) accompanied by modest ruffling and saddle deformations of the C20N4 least-squares plane. The miss shape doming structure is indicated by the values of the displacements from the nitrogen atoms of the pyrrole rings toward the Pyrazine axial ligand.

Solid-state structures of [Cd(TMPP)(Pyz)] complex (**1**) were stabilized through significant non-covalent intermolecular interactions (NCI), such as C-H···N and C-H···Cg H-bonding. Within the crystal structure of (**1**) (Figure 3), the [Cd(TMPP)(Pyz)] (**1**) molecules are linked through weak non-conventional intermolecular H-bond interactions. Thus, the N6 nitrogen atom of the pyrazine axial ligand of the first molecule was involved in a C-H···N H-bond with the H65 hydrogen atom of the pyrazine axial ligand of the second molecule with a distance of 2.962 (7) Å. The H48A hydrogen atom of the methoxy group of the first molecule was also involved in another C-H···N H-bond with the N6 nitrogen atom of the third molecule with a length of 2.676 Å (Figure 3). In addition to this, the 3D crystal packing was reinforced by several weak C-H···π interactions between the H38 hydrogen atom of the methoxy group of one molecule and the pyrrole centroid Cg1 of another molecule with a C38-H38···Cg1 distance of 3.661(3) Å. The same interaction was also noted between the H53 hydrogen atom of the phenyl ring of one molecule and the pyrrole centroid Cg4 of another molecule (C53-H53···Cg1, 3.639(4) Å), (Appendix A). The details of the intermolecular interactions present in complex (**1**) are given in Appendix A.

Notably, the supramolecular material contained 882.58 Å^3^ void space (i.e., ~17.6% of the unit cell volume was located along the a and the b axis) as shown in Figure 4.

### 2.2. Non-Covalent-Interactions (NCI) and Hirshfeld Surface for Intermolecular Systems

Figure 5 and Figure 6 show the intermolecular interaction topology network, where the same color denotes interacting or neighboring molecules. These colors correspond to the symmetry operations (Sym op) in Appendix A, where R is the distance between molecular centroids (mean atomic position). The topology of the intermolecular interactions within the crystal is influenced by the vector properties of each interacting molecule [28]. The energy decomposition by perturbation indicates that the total energy (*E***_tot_**) can be denoted by Equation (1):*E*_tot_ =*E*_ele_ + *E*_pol_ + *E*_dis_ + *E*_rep_(1)
where *E*_el**e**_, *E*_po**l**_, *E*_dis_ and *E*_rep_ represent electrostatic, polarization, dispersion, and repulsive energies, respectively [29].

Other than the two interacting molecules at R = 17.07 Å between the molecular centroids, strong interactions between molecular pairs were observed in the H_2_-TMPP crystal. In general, stronger interactions were observed for the [Cd(TMPP)(Pyz)] crystal, as shown in Appendix A.

The normalized contact distance (d_norm_) consisting of d_e_ and d_i_ is denoted by d_norm_ = d_i_-$r_{i}^{vdw}$/$r_{i}^{vdw}+$ d_e_-$r_{e}^{vdw}$/$r_{e}^{vdw}$, where $r_{i}^{vdw}$ and $r_{e}^{vdw}$ are the van der Waals radii of the atoms. A negative d_norm_ results in a shorter intermolecular contact than *r^vdw^*. Conversely, a positive d_norm_ results in a longer *r^vdw^*. White, red, and blue color isosurfaces are usually revealed by d_norm_, with bright red spots highlighting shorter contacts, white areas revealing contacts around van der Waals separation, and blue regions representing no contacts [30]. It was observed that the [Cd(TMPP)(Pyz)] (**1**) had more sites of intermolecular interaction than the free ligand (Figure 7). These interactions resulted from the C-H···π, C···O and N···O weak contacts, and C-H···N-H between two pyrazine axial ligands, as observed earlier, as well as the O···H hydrogen bonds (Figure 8). It was observed that the C-H···π dominated in both the free ligand and the complex (Table 2).

### 2.3. Natural Bond Orbital Analysis (NBO)

We have considered our metal complex Cd(TMPP) for the interaction with pyrazine (Pyz) ligand. We studied the second-order perturbation energies (SOPE) using the following equation:$$E_{\left(2\right)}=\Delta E_{ij}=q_{i}\frac{{\left(F_{ij}\right)}^{2}}{\Delta \varepsilon }$$
where *qi* and ∆*ε* are the occupancies of the orbital (*i*) and the difference between the energies of the donor orbital (*i*) and an acceptor orbital (*j*), respectively. *F_ij_* is the off-diagonal matrix element.

The table below (Table 3) includes the most relevant *E*_(2)_ energies for the considered complex **1**, as shown in Figure 9. Estimation of those energies was made by NBO single point calculations at the B3LYP-D3/3-21G level of theory. The natural charges analysis was also conducted using the NBO method. For the Cd atom on complex **1**, we noticed a decrease in the natural charge of Cd by 0.1 e when the pyrazine (Pyz)was connected to Cd by its N atom. The nitrogen atom on the Pyz moved from −0.43 e on the ligand alone to −0.51 e on the ligand connected to the Cd complex.

Estimation of the hyperconjugative interactions showed that the ligand nonbonding to metal antibonding orbital ($n_{L}\overset{}{\to }n_{M}^{*}$) were the most significant interactions (Table 3). Complex **1** showed higher E_(2)_ values for all bonds with Cd-N (N1, N2, N3, and N4 were relative to complex **1,** and N5 was relative to the pyrazine). Cd-N1, Cd-N2, Cd-N3, and Cd-N4 had a sum of E_(2)_ equal to 380.62 kcal/mol with relatively close values for complex **1,** with and without pyrazine. When the pyrazine was connected to complex **1**, we noticed higher energy (429.47 and 62.96 kcal/mol for all bonds and N5 (Pyz), respectively). This clearly showed the bond formation between complex **1** and pyrazine via the Cd-N5 bond.

### 2.4. IR and ^1^H NMR Spectroscopies

Experimental and theoretical IR spectroscopy was investigated (Appendix A) to identify the characteristic bands of [Cd (TMPP)(Pyz)] (**1**). In the IR spectrum (Appendix A) of complex (**1**), the band in the range [2956–2902 cm^−1^] is attributed to the stretching frequency C-H of the H_2_-TMPP. The C-H stretching frequency of the pyrazine axial ligand is also observed and is located at 2833 cm^−1^. The C-O stretching band of the methoxy group and the C=C vibrations of the porphyrinic macrocycle show an absorption band at 1243 cm^−1^ and 1510 cm^−1^, respectively. The δ(CCH) bending mode of the TMPP porphyrin tetradentate ligand shows an absorption band at 997 cm^−1^. The experimental data agree with the theoretical data (see Appendix A).

^1^H NMR spectroscopic studies were carried out to obtain further insights into the structure of complex (**1**). The calculated ^1^H NMR of the prepared complex, using the GIAO method was simulated at the B3LYP-D3/6-31G(d) level of theory and compared to the experimental data (Appendix A). The ^1^H NMR experimental spectrum of (**1**) in Appendix A shows a signal at δ = 8.85 ppm assigned to the eight β-pyrrolic protons. Aromatic protons (Ho-Ph/Hm-Ph) resonated in the region between 8.12 and 8.09 ppm. The singlet at 4.10 ppm corresponded to the 12 protons of the methoxy groups (OCH_3_). The absence of the proton signal from the NH pyrrole moieties at δ = −2.73 ppm verified the insertion of the Cd(II) metal ion in the porphyrin cavity and the formation of [Cd(TMPP)] complex. For the Cd(II)-coordinated porphyrin complex (**1**), the characteristic signals of the axial pyrazine ligand were observed at 7.34 ppm (Appendix A). The experimental chemical shift values for Hβ-pyr, p-OCH_3_, and H_(Pyz ligand)_ agrees with the theoretical values, while the Ho-Ph/Hm-Ph chemical shift deviates by over 2 ppm.

### 2.5. Photophysical Properties

#### 2.5.1. UV-Visible Absorption

The UV-Vis data of H_2_-TMPP, [Cd(TMPP)], and [Cd(TMPP)(Pyz)] complex (**1**) were collected in dichloromethane at a concentration of 10^−5^ M. All our synthetic porphyrin species with a selection of related compounds are gathered in Appendix A and compared to several other meso-porphyrin-cadmium metalloporphyrins. As shown in Figure 10, having a Soret and four Q bands is a characteristic of UV-Vis absorption of the H_2_-TMPP free porphyrin (the Soret or B band at 422 nm, and four weaker Q bands between 518 and 650 nm). When the Cd (II) ion occupied the porphyrin cavity, a remarkable bathochromic shift of around 15 nm compared to that of the free ligand occurred, and three Q bands at 437 nm, 573 nm, and 617 nm instead of four were observed. This can be explained by the change in the symmetry from D_2h_ in free-base porphyrin to D_4h_ in the two corresponding cadmium metalloporphyrins [Cd(TMPP)] and [Cd(TMPP)(Pyz)] (**1**). On the other hand, according to the UV-Vis absorption spectra (Figure 10) for [Cd (TMPP)(Pyz)] (**1**), the Soret and Q bands were observed at 440 nm, 579, and 623 nm, respectively. A slight redshift of the Soret and Q bands for complex (**1**) was observed compared to [Cd(TMPP)], which could be related to the increase in the doming distortion caused by the coordinated pyrazine axial ligand [21]. Notably, the important redshift of the absorption bands due to the decrease in the HOMO-LUMO energy was related to the deformation of the porphyrin core [31].

The energy difference between the levels of the HOMO and the LUMO orbitals, which corresponds to the optical energy gap (Eg-op) was determined from the absorption spectrum (UV-Vis) by the Tauc plot method [32]. The Eg-op value was 2.23 eV (Appendix A) which was close to the related Cd(II) metalloporphyrin, and the as-prepared (**1**) was in the semiconductor range.

#### 2.5.2. Fluorescence and Singlet Oxygen Studies

Using an excitation wavelength of 437 nm, fluorescence properties of the meso-methoxyphenyl porphyrin H_2_-TMPP, [Cd(TMPP)] and [Cd(TMPP)(Pyz)] complex (**1**) were registered in CH_2_Cl_2_ solvent at a concentration of 10^−6^ M (Figure 11a). Appendix A recapitulated the fluorescence maxima, the Q(0, 0) and Q(0, 1) bands, the quantum yields (Φf), and the lifetime (τ_f_) of all our synthetic porphyrin species in addition to other designated species. Upon excitation at 437 nm, the free-base porphyrin H_2_TMPP displayed two fluorescence peaks at 656 and 722 nm for Q(0, 0) and Q(0, 1), respectively while for the [Cd(TMPP)], the emission spectrum showed a peak Q(0, 0) at 618 nm and a peak Q(0, 1) at 656 nm. These peaks were blue shifted by 38 nm (from 656 to 618 nm) compared to the H_2_-TMPP free-base porphyrin for Q(0, 0), and by 70 nm (from 722 to 652 nm) for Q(0, 1), followed by a decrease in the fluorescence intensity. For [Cd(TMPP)(Pyz)] (**1**), the fluorescence profile was very similar to the [Cd(TMPP)], and the axial coordinated pyrazine showed a slight perturbation effect on the complex. This was mainly due to the heavy-atom effect on the Cd center [33], which subsequently distorts the porphyrin ring. Additionally, the pyrazine axial ligand encourage intersystem crossing to the triplet state [34,35], resulting in reduced e quantum yields (Φf), which diminished from 0.084 for H_2_TMPP to 0.06 for [Cd(TMPP)], and 0.03 for [Cd(TMPP)(Pyz)]. The (Φf) of (**1**) was higher than that of [Cd(TMPP)(DABCO)] (Φf = 0.01) [21] and lower than that of [Cd(TClPP)(morph)] [36] (Appendix A). The fluorescence lifetime also decreased from 7.8 ns for H_2_TMPP to 1.6 ns for complex (**1**). For the latter, the emission intensity was considerably lower than that of [Cd(TMPP)] and H_2_TMPP, indicating the strong suppression of the recombination of photoinduced carriers in this complex, which played a significant role in enhancing the photocatalytic activity [37].

The singlet oxygen production spectra of our porphyrin species are shown in Figure 11b, giving the maximum emission value at 1270 nm. The calculated values of the singlet oxygen quantum yield (Φ_Δ_) in CH_2_Cl_2_ for the H_2_-TMPP, [Cd(TMPP)], and [Cd(TMPP)(Pyz)] (**1**) are presented in Appendix A. These results show a decrease in Φ_Δ_ values in the order H_2_TMPP (0.73)>[Cd(TMPP)] (0.57)>[Cd(TMPP)(Pyz)] (**1**) (0.13). Thus, metalation of H_2_- TMPP with Cd(II) metal and the axial coordination of the pyrazine led to a reduction in the Φ_Δ_ value compared to that observed for H_2_-TMPP. Previous investigations into Cd(II)-porphyrins have determined that the Cd(II) metal ion is larger than the central hole of the porphyrinato core, which generates an important deformation on the macrocycle [22,33]. Additionally, as reported before, the coordination of axial ligands with a metalloporphyrin can generate a deformation on the macrocycle. Certainly, the axial coordination of the pyrazine molecule to the Cd metal ion would increase the non-planarity. In this case, porphyrin distortion and the heavy atom effect will result in a decrease in the Φ_Δ_ value [35]. On the other hand, as previously revealed, porphyrin metal complexes are a kind of superior light-harvesting material and can generate active species ^1^O_2_ [38,39,40]. This characteristic makes them strong candidates for photocatalytic degradation of organic dye pollutants. Herein, the photocatalytic activity of as-prepared pyrazine-Cd-porphyrin was tested by measuring the degradation of Levafix Blue CA reactive dye. The latter is extensively used in the textile industry and resistant to biodegradation. Degradation of Levafix Blue CA reactive dye into simple molecules can reduce environmental pollution.

### 2.6. Photocatalytic Activity

The photocatalytic activity of the as-prepared [Cd(TMPP)(Pyz)] (**1**) towards the degradation of Levafix Blue CA reactive dye in an aqueous solution under solar light irradiation was evaluated. In each experiment, 20 mg of H_2_TMPP, [Cd(TMPP)], and [Cd(TMPP)(Pyz)] (**1**) catalysts were dispersed in 40 mL of a 10 mg/L solution of Levafix Blue CA reactive dye. Before irradiation, the dispersion was kept in the dark overnight to establish an adsorption/desorption equilibrium. The photocatalytic decomposition of Levafix Blue CA reactive dye was monitored by a decrease in the characteristic absorption band at λ_max_ = 610 nm with the increase in irradiation time in the degradation process (Figure 12a). Furthermore, the concentration changes in Levafix Blue CA reactive dye aqueous solution was plotted against irradiation time (Figure 12b). When no photocatalyst was added, the change in dye concentrations was negligible during the whole irradiation time. The meso-porphyrin H_2_-TMPP induced a 40% photodegradation of Levafix Blue CA reactive dye after 150 min irradiation, while the photodegradation reached 80% with the [Cd(TMPP)(Pyz)] complex (**1**). These results are attributable to the roles of cadmium metal and the pyrazine axial ligand in enhancing catalytic performances.

The reusability of [Cd(TMPP)(Pyz)] (**1**) for photocatalytic degradation of Levafix Blue CA was evaluated. As shown in Appendix A, after five cycles, the photodegradation efficiency decreased from 80% to 60%. This result shows that the photocatalytic activity has good repeatability and considerable stability under the present conditions. The experiment suggest that complex (**1**) is a highly efficient, excellent reusable photocatalyst for the degradation of this organic dye. To test the influence of catalyst concentration in the degradation of Levafix Blue CA dye, a study with three different masses of [Cd(TMPP)(Pyz)] (**1**) was carried out. As shown in Appendix A, an increase in the photocatalyst mass of (**1**) from 10 to 20 to 40 mg in a 10 mg/L of Levafix Blue CA reactive dye solution induced a gradual increase in the degradation kinetics. However, a slight decrease was noted when moving from 20 mg to 40 mg for the photocatalyst mass of (**1**). These results highlight that increasing the amount of photocatalyst used will not speed up the reaction rates beyond a certain point.

The mechanism for the above photocatalytic degradation reaction is proposed in Figure 13. Upon simulated solar light irradiation, electrons (e−) are photogenerated from the valence band (V.B) to the conduction band (C.B) of the catalyst. These e− are then transferred to minimize photocatalytic performances. The photogenerated electrons reduce (e−) O_2_ to O_2_^•^−, which reacts with water to produce hydroxyl radicals (·OH). Additionally, the interaction of holes (h+) with hydroxyl (OH^−^) may also produce hydroxyl radicals ^•^OH and singlet oxygen (^1^O_2_) [41]. Highly reactive oxygen species, ^•^OH, O_2_^•−^ and ^1^O_2_ radicals oxidize the Levafix Blue CA reactive dye into the water, carbon dioxide, and inorganic salts.

## 3. Materials and Methods

### 3.1. Instrumentation

The microwave reactions were performed using a Milestone Microwave Laboratory. H_2_-TMPP purification was performed by flash chromatography using Geduran 60H Silica Gel (63–200 mesh). A Bruker 300 Ultra shield spectrometer (Nancy, France) was used to perform ^1^H NMR spectra. Deuterated DMSO with residual peaks at δ = 2.5 ppm was chosen for NMR experiments at room temperature (T = 298 K), and chemical shifts (δ) were represented in parts per million (ppm). The multiplicity was defined as s and d for singlet and doublet, respectively. Elemental analysis was made by Carlo Erba model 1106 microanalyzer (INEOS RAS). Fourier-transformed IR spectra were recorded on a PerkinElmer Spectrum. UV-vis spectra of porphyrin derivatives were measured with a Shimadzu UV-3600 spectrophotometer. The emission spectra of porphyrin derivatives were recorded at room temperature with a Fluorolog-3 spectrofluorometer. The meso-tetraphenyl porphyrin H_2_-TMPP was employed as fluorescence quantum yield standard (Φ_f_ = 0.11) in a toluene solution [42]. Singlet oxygen quantum yield (Φ_∆_) was determined by direct measurement of the infrared luminescence using H_2_-TMPP as mentioned above, with the ^1^O_2_ quantum yield being (Φ_∆_ = 0.63) [43].

### 3.2. Crystallography

A single crystal of complex (**1**) was mounted on glass fibers, and intensity data were collected at 150 K using a Bruker-AXS APEXII diffractometer equipped with a graphite monochromatic Mo-Kα (λ = 0.71073 Å) radiation. The multi-scan method was used for absorption corrections [44]. The structure was solved by direct methods by using SIR-2004 [45] and refined with the full-matrix least-squares method based on F2 (SHELXL-2014) [46]. The PLATON SQUEEZE procedure was applied to remove the disordered solvents [47]. Least square refinements with anisotropic thermal motion parameters for all non–hydrogen atoms and isotropic measurements for the remaining atoms were employed. The images were produced using MERCURY software [48]. CCDC 2069722 for compound (**1**) contains the supplementary crystallographic data for this paper. These data can be obtained free of charge from the Cambridge Crystallographic Data Centre via www.ccdc.cam.ac.uk/data_request/cif (accessed on 11 March 2021).

### 3.3. Computational Methods

The calculations in the present studies were performed using Gaussian09 software [49]. Model preparation and analyses were conducted through Gaussview [50]. The density functional theory (DFT) method was applied using the hybrid functional (B3LYP) [51] along with a split-valence double-zeta basis set (namely 6-31G). The basis set includes five d-type Cartesian-Gaussian polarization functions on all light atoms (C, N, O and H) [52]. For weak interactions, dispersion correction was considered in the energy evaluations using the Grimme D3 method [53]. Geometry optimizations with no symmetry constraints were applied at B3LYP-D3/6-31G(d) level of theory. The geometries were checked to ensure they had the minimum potential energy structures by calculating vibrational frequency calculations. Natural bond orbital (NBO) analysis was conducted to estimate the delocalization interactions of our complex at B3LYP-D3/3-21G level of theory [54,55,56,57]. Hirshfeld analysis was performed using crystal explorer software by analyzing the interacting sites on the central molecule with a neighboring molecule. This was followed by measuring the percentage contribution between atoms by ascertaining the distances between the surface and the nucleus internal and external to the surface with no property at a high standard resolution. This was followed by measuring the interacting energies between pairs of molecules within the crystal’s unit cell at the B3LYP/DGDZVP level of theory [58].

### 3.4. Photocatalytic Measurements

The photocatalytic activity of the prepared [Cd(TMPP)(Pyz)] (**1**) was evaluated by the degradation of a Levafix Blue CA organic dye under solar light. A 20 mg portion of catalyst was added to the pollutant solution (10 mg/L). Before the irradiation, the mixture was stirred at room temperature for 90 min in the dark to ensure adsorption–desorption equilibrium. The solution was then stirred continually under solar light. Approximately 4 mL of solution suspensions were taken out and separated by centrifugation. The UV−vis spectrum of the resulting solution was measured. The Blue CA dye showed a characteristic peak at (λ = 610 nm). It was employed to monitor the photodegradation process. The percentage of degradation was determined using the following equation:Dye degradation (%) = [(C_0_ − C)/C_0_] × 100
where C_0_ is the initial concentration of dye solution at 0 min and C is the concentration of dye solution at a certain time.

### 3.5. Synthesis

All reagents employed were commercially available and were used without further purification. Solvents were appropriately distilled and dried before used.

The penta-coordinated Cd^II^ porphyrin complex (**1**) was prepared following three steps with an 85% overall yield (Figure 2). The preparation of the meso-tetrakis(para-methoxyphenyl) porphyrin H_2_-TMPP and the metalation with cadmium(II) (Figure 2, steps (i) and (ii), respectively) was performed using the microwave-assisted method described in the literature [59,60]. The coordination of pyrazine to the cadmium metal center was performed in CHCl_3_ (Figure 2, step (iii)).

#### 3.5.1. Synthesis of Meso-Tetrakis(para-methoxyphenyl) Porphyrin, H_2_-TMPP

A mixture of 4-methoxybenzaldehyde (1090 mg, 8.0 mmol), pyrrole (537 mg, 8.0 mmol) in propionic acid (2 mL) was placed in a cylindrical vessel and deposited in a Milestone Microwave. The mixture was irradiated for 15 min at 120 °C and 680 W. After cooling, the reaction mixture was dissolved in CHCl_3_ (15 mL) and washed with water (35 mL). After separation, the water phase was additionally extracted with CHCl_3_ (50 mL). The combined organic layers were washed with water (150 mL), dried over MgSO_4_, and concentrated *in vacuo*. The crude product was purified by flash column chromatography using CHCl_3_/hexane (1/1, *v*/*v*) as an eluent to afford H_2_-TMPP (397 mg, 27%) purple solid. Anal. data for C_48_H_38_N_4_O_4_ calcd. (%): C, 78.45; H, 5.21; N, 7.62, found (%): C, 78.40; H, 5.19; N, 7.58. Melting point: >300 °C. UV-Vis [λ_max_ in solvent CH_2_Cl_2_ (nm), ε × 10^−3^ (mol^−1^·L·cm^−1^)]: 422(390.2), 518(14.10), 555(9.5), 594(3.9), 650(4.2). FTR-IR (solid, cm^−1^): 3321 ν(NH), 2930–2829 ν(CH), 1503 ν(C=C), 1246 ν(OCH_3_), 990 δ(CCH). ^1^H NMR (300 MHz, DMSO-d6, 298 K): δ(ppm) −2.73 (s. 2H, NH_pyr_), 4.11 (s, 12H, OCH_3_), 8.09 (s, 8H, Hm-Ph), 8.12 (s, 8H, Ho-Ph), 8.85(s, 8H, H_β-pyr_).

#### 3.5.2. Synthesis of Meso-Tetrakis(para-methoxyphenyl)porphyrinato)-cadmium(II), [Cd(TMPP)]

H_2_-TMPP (100 mg, 0.14 mmol) and CdCl_2_·2H_2_O (300 g, 1.36 mmol) in DMF (30 mL) were irradiated in a Pyrex bottle using a microwave oven for 30 min at 130 °C and 480 W. The solution changed from purple to green and the reaction was measured by UV-vis spectroscopy. The reaction mixture was then filtered, evaporated to dryness and afforded [Cd(TMPP)] as dark green solid (0.45 g, 87%). Anal. data for C_48_H_36_CdN_4_O_4_ calcd. (%): C, 68.20; H, 4.29; N, 6.63, found (%): C, 68.36; H, 4.32; N, 6.74. Melting point: 251 °C. UV-Vis [λ_max_ (nm) in CH_2_Cl_2_, (ε × 10^−3^, mol^−1^ L cm^−1^)]: 437(125), 573(4.2), 617(2.4). FTR-IR (solid, cm^−1^): 2986–2836 ν(NH), 1601 ν(C=C), 1507 ν(C=N), 1329 ν(CN), 1246 ν(OCH_3_), 999 δ(CCH). ^1^H NMR (300 MHz, DMSO-d6, 298 K): δ (ppm) 4.10 (s, 12H, OCH_3_), 8.11 (s, 8H, Hm-Ph), 8.14 (s, 8H, Ho-Ph), 8.86(s, 8H, H_β-pyr_).

#### 3.5.3. Synthesis of the (Pyrazine)[Meso-tetrakis(para-methoxyphenyl)-porphyrinato)-cadmium(II), [Cd(TMPP)(Pyz)] (**1**)

An excess amount of pyrazine (60 mg, 0.43 mmol) was added to a solution of [Cd(TMPP)] (20 mg, 0.024 mmol) in CHCl_3_ (5 mL). The solution was stirred overnight at room temperature before being filtrated. Crystals of complex **1** were obtained by slow diffusion of hexane into the CHCl_3_ solution (17 mg, 85%). Anal. data for C_52_H_40_CdN_6_O_4_ calcd. (%): C, 67.49; H, 4.35; N, 9.08, found (%): C, 67.65; H, 4.38; N, 9.17. Melting point: 254 °C. UV-Vis [λ_max_ (nm) in CH_2_Cl_2_, (ε × 10^−3^, mol^−1^ L cm^−1^)]: 440(113), 579(15.2), 623(3.2). FTR-IR (solid, cm^−1^): 2956–2902 ν(CH)porph, 2833 ν(CH) pyz., 1510 (ν(C=N), 1243 ν(OCH_3_), 997 δ(CCH). ^1^H NMR (300 MHz, DMSO-d6, 298 K): δ (ppm) 4.05 (s, 12H, OCH_3_), 7.34 (d, *J* = 9 Hz, 4H, H_pyz_), 8.08 (s, 8H, Hm-Ph), 8.10 (s, 8H, Ho-Ph), 8.73 (s, 8H, H_β-pyr_).

## 4. Conclusions

We reported here the synthesis of a novel (Pyrazine)(meso-tetrakis(4-tert-methoxyphenyl)-porphyrinato)-cadmium(II) coordination compound, abbreviated as [Cd(TMPP)(Pyz)] (**1**). This new penta-coordinated Cd^II^ porphyrin complex was structurally characterized by X-ray diffraction crystallography. The structure of complex (**1**) revealed that the Cd(II) metal ion was located out of the 24-atom porphyrinato core mean plane, inducing a significant porphyrinato core doming distortion. The supramolecular arrangement was maintained by C-H···N intramolecular H-bonds and weak C-H⋯Cg π intermolecular interactions. Using crystal explorer software at the B3LYP/DGDZVP basis set, [Cd(TMPP)(Pyz)] revealed stronger intermolecular interactions than H_2_-TMPP. In addition to this, it was also revealed that the C-H···π formed the strongest interactions in both H_2_-TMPP and [Cd(TMPP)(Pyz)] compounds. Additionally, the photophysical properties of H_2_-TMPP, [Cd (TMPP)], and [Cd(TMPP)(Pyz)] (**1**) were reported. The UV-visible spectra revealed bathochromic shifts for the Soret and Q absorption bands associated mainly with the dome distortion of the porphyrinato macrocycle. However, the fluorescence and the singlet oxygen values of the free-base porphyrin H_2_-TMPP were higher than the Cd(II) complexes due to the heavy-atom effect and the distortion of the porphyrin core on the latter. DFT calculations on complex (**1**) were carried out, and the results agrees with experimental data. NBO analysis was carried out to determine the atomic charge distribution of the prepared compound and to identify how the Cd-porphyrin and the pyrazine axial ligand interacted with each other. The efficient catalytic ability of complex (**1**) for the photodegradation of Levafix Blue CA reactive dye under solar light irradiation indicated the potential application of this complex both in environmental and industrial domains.

## Data Availability

Data are contained within the article or Appendix A.

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
