# Peer review of "Synthesis, Characterization, DFT and Photocatalytic Studies of a New Pyrazine Cadmium(II) Tetrakis(4-methoxy-phenyl)-porphyrin Compound"

_molecules, 2022, doi:10.3390/molecules27123833_

Round 1

Reviewer 1 Report

The authors describe the synthesis and investigation of the photophysical property of porphyrin cadmium complex. The obtained porphyrin was characterized by NMR, FT-IR and X-ray crystallography.In addition, the cadmium porphyrin can work as the photocatalysis under photoirradiation to decompose the Levafix Blue. The findings in this manuscript will be a good guidance of development of porphyrin based photocatalysis. However, there are many careless mistakes and questions to this paper. Therefore, in my opinion, it is suitable for publication in Molecules, after major revision.

[1] As mentioned above, there are a lot of mistakes in this paper. I recommend the author should carefully pay attention to notations such as abbreviation name of the cadmium porphyrin, superscripts and subscripts in the units, and italics for physical quantity.

[2] In Figures 5 and 6, these figures are very complicated. It is hard to understand the distances and intermolecular interactions. It should be redrawn in an easy-to-understand.

[3] In the 1H NMR, assignment of signals of the cadmium porphyrin is not current. The numbers of peaks are not match between Figure S10 and experimental section. Additionally, I cannot agree the theoretical values are similar to experimental data because the peak position of meso-aryl groups are different over 2 ppm between theoretical and experimental values.

[4] After coordination of pyrazine to the cadmium porphyrin, the peaks positions of UV-vis absorption are red-shifted. The authors explained the reason of this phenomena by an increases of the p-conjugation. However, the pyrazine unit is located at the almost perpendicular on porphyrin. In addition, your previous paper about DABCO coordinated cadmium porphyrin is reported that the reason of red-shifted the absorption peaks is the deformation of the porphyrin by coordination. You should explain which effects are dominated to the absorption peaks of pyrazine coordinated cadmium porphyrin.

[5] In the emission spectra, I feel strange the emission peaks of freebase porphyrin and cadmium porphyrin are the same at 656 nm. I suspect that freebase porphyrin is contaminated into the cadmium porphyrin or is demetallation during measurement. Therefore, excitation spectrum should be considered.   

[6] For photocatalytic activity, why the compound 1 works well compared with freebase and normal cadmium porphyrin, although the compound 1 showed the smallest singlet oxygen production efficiency.

Author Response

            Response to Reviewers

Dear Prof Ms. Alysa Zhao
Assistant Editor

We would like to thank you for considering our manuscript” Synthesis, Characterization, DFT and Photocatalytic studies of a new Pyrazine Cadmium(II) Tetrakis(4-methoxy-phenyl)-porphyrin compound” for publication in Molecules.

we appreciate the opportunity to resubmit our manuscript after revising it according to the reviewers’ suggestions, which we have addressed in full below. Please find our comments marked up in blue and changes in the manuscript highlighted in red.

 Best regards,

Chadlia Mchiri on behalf of all authors.

 We thank the reviewers for their careful reading and valuable suggestions, which led to a more concise presentation of our results.

Reviewer #1

The authors describe the synthesis and investigation of the photophysical property of porphyrin cadmium complex. The obtained porphyrin was characterized by NMR, FT-IR and X-ray crystallography. In addition, the cadmium porphyrin can work as the photocatalysis under photoirradiation to decompose the Levafix Blue. The findings in this manuscript will be a good guidance of development of porphyrin based photocatalysis. However, there are many careless mistakes and questions to this paper. Therefore, in my opinion, it is suitable for publication in Molecules, after major revision.

 Request

[1] As mentioned above, there are a lot of mistakes in this paper. I recommend the author should carefully pay attention to notations such as abbreviation name of the cadmium porphyrin, superscripts and subscripts in the units, and italics for physical quantity.

Response:

We thank the reviewer for mentioning these typos mistakes. We revised the manuscript and corrected these mistakes.

Request

[2] In Figures 5 and 6, these figures are very complicated. It is hard to understand the distances and intermolecular interactions. It should be redrawn in an easy-to-understand.

Response:

Figures 5 and 6 were updated in the revised manuscript.

Request

[3] In the 1H NMR, the assignment of signals of the cadmium porphyrin is not current. The numbers of peaks are not match between Figure S10 and experimental section. Additionally, I cannot agree the theoretical values are similar to experimental data because the peak position of meso-aryl groups are different over 2 ppm between theoretical and experimental values.

Response:

The reviewer is right, we made the corrections in the experimental section of the revised manuscript and in table S8.

Also we agree with the referee that the theoretical values are not similar to the experimental, we correct this mistake in section 2.4. IR and 1H NMR spectroscopies, and we add the following paragraph.

“The experimental chemicals shift values of  Hβ-pyr, p- OCH3, and  H(Pyz ligand) are close to the theoretical values. While for Ho-Ph/ Hm-Ph there is a difference over 2 ppm between theoretical and experimental values”.

Request

[4] After coordination of pyrazine to the cadmium porphyrin, the peaks positions of UV-vis absorption are red-shifted. The authors explained the reason of this phenomena by an increase of the p-conjugation. However, the pyrazine unit is located at the almost perpendicular on porphyrin. In addition, your previous paper about DABCO coordinated cadmium porphyrin is reported that the reason of red-shifted the absorption peaks is the deformation of the porphyrin by coordination. You should explain which effects are dominated to the absorption peaks of pyrazine coordinated cadmium porphyrin.

Response:

Thank you for this remark

The red shift of the absorption bands is related in the majority to the significant deformation of the porphyrin core caused at first to the Cd(II) ion which is too large to fit into the central hole of the TMPP, which generates significant deformation of the TMPP, and the coordination of pyrazine axial ligand which tends to increase the doming distortion by pulling the Cd(II) metal ion out of the porphyrinato plane (see Crystallographic Section).

We removed the sentence “which indicates increased π-conjugation upon and opening up opportunities to develop optoelectronic materials “and we change by the new one:

 “which could be related to the increase in the doming distortion caused by the coordinated pyrazine axial ligand [21] “

Request

[5] In the emission spectra, I feel strange the emission peaks of freebase porphyrin and cadmium porphyrin are the same at 656 nm. I suspect that freebase porphyrin is contaminated into the cadmium porphyrin or is demetallation during measurement. Therefore, excitation spectrum should be considered.  

Response: In Fluorescence and Singlet oxygen studies section the following paragraph is written :

“Upon excitation at 437 nm, the free base porphyrin (H2TMPP) displayed two fluorescence peaks at 656 and 722 nm for Q(0,0) and Q(0,1), respectively. Compared to H2TMPP, the [Cd(TMPP)] emission spectrum gives, from 656 to 618 nm, a 34 nm and from 722 to 652 nm, a 63 nm blue-shifts in the same Q bands, respectively, followed by a decrease of the fluorescence intensity”.

which would be meant that the free base porphyrin show two peaks Q(0,0) at 656 nm and Q(0,1) at 722 nm. Where, the cadmium porphyrin [Cd(TMPP)] display also two peaks  Q(0,0) at 618 nm and Q(0,1) at 652 these peaks are blue-shifted compared to those of free base porphyrin. It seems not clear we removed and we replaced with this paragraph:

“Upon excitation at 437 nm, the free base porphyrin (H2TMPP) displayed two fluorescence peaks at 656 and 722 nm for Q(0,0) and Q(0,1), respectively. While for the [Cd(TMPP)] the emission spectrum showed a peak  Q(0,0)  at 618 nm, and a peak Q(0,1) at 656 nm. These peaks are blue-shifted compared to H2TMPP freebase porhyrin with 38 nm (from 656 to 618 nm) for Q(0,0), and a 70 nm (from 722 to 652 nm ) for Q(0,1) respectively, followed by a decrease of the fluorescence intensity.”

Request

 [6] For photocatalytic activity, why the compound 1 works well compared with freebase and normal cadmium porphyrin, although the compound 1 showed the smallest singlet oxygen production efficiency.

Response: It is true that free porphyrin produces much more singular oxygen, on the other hand for Cd porphyrin and complex 1 the fluorescence and singular oxygen decreases, this said probably to the quenching effect of cadmium, on the other hand, previous studies have shown the important influence on the catalytic performance of transition metals and among which cadmium in the photodegradation of dyes, among them is cadmium in the photodegradation of dyes: https://doi.org/10.1016/j.jssc.2020.121493; https://doi.org/10.1016/j.jssc.2019.121168.

In this work the porphyrinic complex of cadmium with pyrazine shows the best result in the photodegradation of Levafix Blue CA, this needs to be studied much more thoroughly by varying the parameters in future research.

Hopefully, our replies and further work could cover and satisfy the referees’ comments. We hope this version is now suitable for publication in Molecules

Sincerely yours,

Dr. Chadlia Mchiri

Reviewer 2 Report

  1. For the structures of [Cd (TMPP)] and [Cd(TMPP)( Pyz)] (1), how toconfirm that the reaction is free of dimers or other isomers about another N atom? only for weak interactions?
  2. In the text, only penta-coordinated CdII porphyrin complex (1)was discussed,how to evaluate the calculation results and other properties, with there is no comparability.
  3. The HF/3-21G basis setis simple for the crystal and you can slightly improve the basis set to calculate.
  4. 5 and Fig.6 are blurry.
  5. For NBO, some new literature should be referenced, such as Molecules 2019, 24, 2090;  2020, 25, 4052.

Author Response

            Response to Reviewers

Dear Prof Ms. Alysa Zhao
Assistant Editor

We would like to thank you for considering our manuscript” Synthesis, Characterization, DFT and Photocatalytic studies of a new Pyrazine Cadmium(II) Tetrakis(4-methoxy-phenyl)-porphyrin compound” for publication in Molecules.

we appreciate the opportunity to resubmit our manuscript after revising it according to the reviewers’ suggestions, which we have addressed in full below. Please find our comments marked up in blue and changes in the manuscript highlighted in red.

 Best regards,

Chadlia Mchiri on behalf of all authors.

 We thank the reviewers for their careful reading and valuable suggestions, which led to a more concise presentation of our results.

Reviewer #2

Request

  1. For the structures of [Cd (TMPP)] and [Cd(TMPP)( Pyz)] (1), how to confirm that the reaction is free of dimers or other isomers about another N atom? only for weak interactions?

Response: For the structure of  [Cd (TMPP)] we don’t have CIF file to confirm that reaction free of dimers. But it is thought that it is difficult to get a dimer, as the dimerization is probably through the axial ligand.

 For the structure of [Cd(TMPP)( Pyz)] (1), according to the X-ray diffraction, it’s confirmed that the unit cell consists of one monomer of [Cd(TMPP)( Pyz)] in the asymmetric. While in the crystal lattice the two molecules of [Cd(TMPP)( Pyz)] are linked via of C-H…N intermolecular interaction between the two Nitrogen atoms of two pyrazine ligand.

Request

  1. In the text, only penta-coordinated CdII porphyrin complex (1) was discussed, how to evaluate the calculation results and other properties, with there is no comparability.

Response:

We thank the referee for this valuable hint, we add the calculation results and we compared with other CdII porphyrin complex as described in the following paragraph:

“also the consequence of distortion of the porphyrin ring next to the metalation with Cd(II), and the addition of the pyrazine axial ligand, which encourages intersystem crossing to the triplet state [34-35] resulting in the decrease of the quantum yields (Φf) which diminish from 0.084 for H2TMPP to 0.06 for [Cd(TMPP)],  and 0.03 for [Cd(TMPP)(Pyz)], the (Φf) of (1) is higher than that of [Cd(TMPP)(DABCO)] (Φf =0.01) [21] and lower than that of [Cd(TClPP)(morph)] [36] (Table S4). The fluorescence lifetime also is decreased from 7.8 ns, for H2TMPP, to 1.6 ns for complex (1).”

Request

  1. The HF/3-21G basis set is simple for the crystal and you can slightly improve the basis set to calculate.

Response: We selected the B3LYP/DGDZVP basis set.

Request

  1. Fig 5 and Fig.6 are blurry.

Response: We decided to run it on a supercomputer rather.

Request

  1. For NBO, some new literature should be referenced, such as Molecules 2019, 24, 2090;  202025, 4052.

Response: We agree with the reviewer. The reference has been added.

Hopefully, our replies and further work could cover and satisfy the referees’ comments. We hope this version is now suitable for publication in Molecules

Sincerely yours,

Dr. Chadlia Mchiri

Reviewer 3 Report

The authors have presented their work on "Synthesis, Characterization, DFT and Photocatalytic studies of 2 a new Pyrazine Cadmium(II) Tetrakis(4-methoxy-phenyl)-por-3 phyrin compound," which might be of interest to readers of Molecules.

The research undertaken and presented in the manuscript is of significance in the field of porphyrin compounds. These are some of the points the authors should address, which might improve their manuscript's scientific soundness and help readers understand the research better.

(1) In the abstract, it would be better to include the singlet oxygen quantum yield and the relative quantum yield for the synthesized porphyrin compound. The authors should also have the synthetic yield of the compound in the introduction.

(2) In the introduction, it would be better for the authors to emphasize the fact that although porphyrins can be extracted from plant matters, the amount obtained is minimal, and therefore synthesizing them will be a better option.

(3) The authors should also clearly mention some of the main applications of porphyrins.

(4) The term heterogeneous photocatalysis should be clearly defined.

(5) In section 2.5.1 the sentence "The redshift of the Soret and Q bands of complex (1) indicates an increase in the π 245 -conjugation and the HOMO–LUMO energy gap [27,28] due to the increase of the molecule symmetry" it is difficult to understand does the author means the homo-lumo energy gap is increasing or decreasing. Please clarify this statement.

(6) The application of the synthesized compound in dye decomposition is noteworthy, but cadmium itself poses toxicity issues. Can the author describe how they are going to address this?

Overall the research presented will be of interest to researchers working with porphyrins.

Author Response

            Response to Reviewers

Dear Prof Ms. Alysa Zhao
Assistant Editor

We would like to thank you for considering our manuscript” Synthesis, Characterization, DFT and Photocatalytic studies of a new Pyrazine Cadmium(II) Tetrakis(4-methoxy-phenyl)-porphyrin compound” for publication in Molecules.

we appreciate the opportunity to resubmit our manuscript after revising it according to the reviewers’ suggestions, which we have addressed in full below. Please find our comments marked up in blue and changes in the manuscript highlighted in red.

 Best regards,

Chadlia Mchiri on behalf of all authors.

 We thank the reviewers for their careful reading and valuable suggestions, which led to a more concise presentation of our results.

Reviewer #3

The authors have presented their work on "Synthesis, Characterization, DFT and Photocatalytic studies of 2 a new Pyrazine Cadmium(II) Tetrakis(4-methoxy-phenyl)-porphyrin compound," which might be of interest to readers of Molecules.

The research undertaken and presented in the manuscript is of significance in the field of porphyrin compounds. These are some of the points the authors should address, which might improve their manuscript's scientific soundness and help readers understand the research better.

Request

  1. In the abstract, it would be better to include the singlet oxygen quantum yield and the relative quantum yield for the synthesized porphyrin compound. The authors should also have the synthetic yield of the compound in the introduction.

Response: Thank you for your remarks, The abstract was revised. All modifications were highlighted in red color.

“Singlet oxygen revealed that the H2TMPP porphyrin appears as the most efficient in singlet oxygen production with a value quantum yield of (FD = 0.73) compared to  [CdTMPP] (FD = 0.57) and [CdII(TMPP)(Pyz)] (1) (FD = 0.13).”

Also the synthetic yield of the compound was added in the introduction.

“This complex (1) is obtained with a yield of around 85%.”

Request

  1. In the introduction, it would be better for the authors to emphasize the fact that although porphyrins can be extracted from plant matters, the amount obtained is minimal, and therefore synthesizing them will be a better option.

Response: Thank you, we added this paragraph in the introduction

“Porphyrins are already existed in nature and called the pigments of life due to their extreme importance among natural compounds [1]. Although porphyrins can be extracted from plant matters, the amount obtained is minimal, and therefore synthesizing them will be a better option, to obtain much better yields and diversity in their structure, characteristics, and functions which make it with a wide range of applications [2-4]”.

Request

  1. The authors should also clearly mention some of the main applications of porphyrins.

Response: We cited some applications of porphyrins in the introduction.

“The structural diversity of porphyrins, with core-, meso-, β-modifications, having a role in donating electrons, suggest a way to tune the electronic structures and their properties as dye sensitizers in DSSCs [5-6]. For example 5-(4-carboxyphenyl)-10,15, 20-tritolylporphyrin (H2TC1PP) and 5,10,15,20-tetrakis(4-carboxy phenyl)porphyrin (H2TC4PP) have been used as photosensitizers in TiO2 DSSCs and investigated as effectual photosensitizers for solar-energy alteration [7]. In addition, porphyrins have important potential applications in Cancer treatment, Thus numerous porphyrin-based inorganic nanoparticles (NPs) have been designed and applied in cancer therapy [8]”.

Request

  1. The term heterogeneous photocatalysis should be clearly defined.

Response: We try to define the term heterogeneous photocatalysis, we added the following paragraph in the introduction

“Heterogeneous photocatalysis is an emerging technique valuable for water and air purification and remediation. In heterogeneous photocatalysis, the photocatalyst is present as a solid with the reactions taking place at the interface between phases, i.e., solid-liquid or solid-gas”.

Request

  1. In section 2.5.1 the sentence "The redshift of the Soret and Q bands of complex (1) indicates an increase in the π 245 -conjugation and the HOMO–LUMO energy gap [27,28] due to the increase of the molecule symmetry" it is difficult to understand does the author means the homo-LUMO energy gap is increasing or decreasing. Please clarify this statement.

Response: We agree with the referee this statement is not clear, we changed by new statement and added references.

Notably, the important red shift of the absorption bands as a result of the decrease of the HOMO-LUMO energy is related to the deformation of the porphyrin core [31].

Request

  1. The application of the synthesized compound in dye decomposition is noteworthy, but cadmium itself poses toxicity issues. Can the author describe how they are going to address this?.

Response: In this study, we have synthesized and characterized cadmium porphyrin and tested this compound as a catalyst in the photodegradation of organic pollutants. It is true that cadmium is toxic but we think that it can be used in the photocatalytic of organic pollutants from water. Even previous studies have shown the performance of different compounds and complexes with cadmium in the photodegradation of organic dyes as shown in the following references:

https://doi.org/10.1016/j.mset.2018.09.002; DOI: 10.1021/acs.cgd.8b01040 ; dx.doi.org/10.1021/cg401601d; DOI:  https://doi.org/10.1002/cplu.201402220

Hopefully, our replies and further work could cover and satisfy the referees’ comments. We hope this version is now suitable for publication in Molecules

Sincerely yours,

Dr. Chadlia Mchiri
